# Operationalising the Family-Friendly Medical Workplace and the Development of FAM-MED, a Family-Friendly Self-Audit Tool for Medical Systems: A Delphi Consensus

**DOI:** 10.3390/healthcare11121679

**Published:** 2023-06-07

**Authors:** Carmelle Peisah, Adrianna Sheppard, Susan Mary Benbow, Alison Loughran-Fowlds, Susan Grayson, Jenny E. Gunton, Anuradha Kataria, Rosalyn Lai, Kiran Lele, Carolyn Quadrio, Danette Wright, Loyola McLean

**Affiliations:** 1Faculty of Medicine and Health, The University of Sydney, Sydney, NSW 2006, Australia; jenny.gunton@sydney.edu.au (J.E.G.); anuradha.kataria@health.nsw.gov.au (A.K.); kiran.lele@health.nsw.gov.au (K.L.); loyola.mclean@sydney.edu.au (L.M.); 2Discipline of Psychiatry and Mental Health, Faculty of Medicine, University of New South Wales, Sydney, NSW 2052, Australia; rosalyn.lai@health.nsw.gov.au (R.L.); carolyn@quadrio.com.au (C.Q.); 3Westmead Hospital, Sydney, NSW 2145, Australia; adrianna.sheppard@health.nsw.gov.au (A.S.); susan.grayson@health.nsw.gov.au (S.G.); 4Centre for Ageing and Mental Health, Faculty of Health and Social Care, University of Chester, Chester CH1 4BJ, UK; drsmbenbow@gmail.com; 5Blacktown Hospital, Sydney, NSW 2148, Australia; alison.loughranfowlds@health.nsw.gov.au (A.L.-F.); danette.wright@health.nsw.gov.au (D.W.); 6Westmead Institute for Medical Research, Sydney, NSW 2145, Australia; 7Royal North Shore Hospital, Sydney, NSW 2065, Australia; 8School of Medicine, Western Sydney University, Sydney, NSW 2751, Australia; 9Westmead Psychotherapy Programme for Complex Traumatic Disorders, Cumberland Hospital, Sydney, NSW 2145, Australia

**Keywords:** family-friendly, medical workplace, doctors’ heath, physicians’ health, occupational stress, work–family conflict

## Abstract

Globally, the call for Family-Friendly (FF) workplaces is loud and clear. However, this call is inaudible in medical workplaces, despite both well-established benefits of FF workplaces across businesses and well-known effects of work–family conflict on the well-being and practice of doctors. We aimed to use the Delphi consensus methodology to: (i) operationalise the Family-Friendly medical workplace and (ii) develop a Family-Friendly Self-Audit tool for medical workplaces. The expert medical Delphi panel was deliberatively recruited to capture a breadth of professional, personal, and academic expertise, diversity of age (35–81), life stage, family contexts and lived experience of dual commitments to work and family, and diversity of work settings and positions. Results reflected the inclusive and dynamic nature of the doctor’s family and the need to adopt a family life cycle approach to FF medical workplaces. Key processes for implementation include holding firms to zero discrimination; flexibility and openness to dialogue and feedback; and a mutual commitment between the doctor and the department lead to best meet the doctor’s individualised needs while still ensuring optimal patient care and team support and cohesion. We hypothesise that the Department Head may be the key to implementation but recognise the workforce constraints to realising these aspirational systemic shifts. It is time we acknowledge that doctors have families, to narrow the gap between identifying as a partner, mother, father, daughter, son, grandparent, and identifying as a doctor. We affirm the right to be both good doctors and good family members.

## 1. Introduction

Globally, the call for Family-Friendly (FF) workplaces is loud and clear [1,2]. We just cannot hear it in medical workplaces. This is ironic given the bi-directional, recursive relationship between doctors and their families with mutual effects not only on health and well-being but also on the practice of medicine [3,4,5,6]. In the 2013 (updated in 2019) Australian National Mental Health Survey of Doctors and Medical Students, comprising a sample of 12,252 doctors, 26.8% of doctors reported “conflict between study/career and family/personal responsibilities”, 18.4% reported caring for a family member, and 15.5% reported death of a family member or close friend in the preceding 12 months [7]. These reported items are only some of the plethora of competing, and often unbearable, work and family responsibilities synonymous with a medical career [8]. These incompatible work and family role pressures, termed “work–family conflict” [9], have profound effects on the well-being of doctors, particularly female doctors [10,11]. The COVID pandemic only fuelled gender inequities imposing disproportionate role burden and role conflict on female doctors, regardless of their particular family context or cultural setting, demonstrated across Canada, Australia, the United States, and Japan [12,13,14,15,16]. Having said that, men who undertake a significant proportion of family caregiving roles will have had similar pressures imposed upon them during COVID and are now rapidly emerging as the new invisible carers in medicine.

The benefits of FF work cultures are well established. Positive outcomes include, but are not limited to, improved parental bonding, child development and health, gender balance at home, eldercare, carer well-being, quality family time, family cohesion and well-being, and reduced family conflict [17]. Hitherto in medicine, FF policies have largely been restricted to academic university-based contexts and particularly driven by litigation in relation to equal opportunity laws, damages under family leave provisions, and extension of tenure clocks rather than family welfare per se. For example, the University of California (UC) first developed Family-Friendly Accommodation Policies for faculty in 1988, with modifications in 2006 [18]. 

However, it is clear that policies themselves do not lead to a change in culture, often because they sit in drawers, unused and unread. Notably, when UC Davis Schools of Medicine and Veterinary Medicine and College of Biological Science faculty members were surveyed, despite large percentages with family care responsibilities, use and knowledge of FF policies were low, albeit higher in women (6.7% of female compared with 0% of male respondents) [19]. Identified barriers to the use of policies included concerns regarding service load and burden on colleagues, financial considerations, fear of repercussions, inability to stop work, especially on grant-funded research, and slower career progress. 

Beyond policy, UC developed a practical Family-Friendly Toolkit to specifically assist department chairs and deans to develop Family-Friendly cultures in their departments [18]. Their FF tips include, but are not limited to: (i) making the department FF a priority and goal; (ii) offering and supporting resources and practices that allow faculty to successfully integrate work and family needs; (iii) reviewing and assessing department FF practices and climate; (iv) rendering unconscious bias around caregiving and gender conscious; (v) highlighting, supporting, and advertising FF policies; (vii) making FF the standard for conducting business, not a special privilege; (vi) zero tolerance for discriminatory or disparaging comments and behaviours, making it clear that these are unacceptable and violate the rules governing professional conduct; (vii) proactive hiring and recruitment including those who have slowed down career for family caregiving reasons; (viii) highlighting departmental FF benefits for new recruits [18]. Family-Friendly Workplaces, UNICEF Australia, have taken this further and developed a benchmarking and certification system based on leading practice guidelines for businesses to develop work and family action plans to embed FF workplace cultures [20]. 

Apart from the aforementioned psychological and well-being outcomes, business clearly recognises the commercial advantages of FF workplaces. These include attraction and retention of employees, better workplace participation, reduced unplanned absenteeism and turnover, higher loyalty and engagement, improved productivity and performance, and reduced legal liability and risk [17,21,22]. With respect to the first of these advantages, contemporary health systems, particularly public health systems, are currently plagued with attrition of doctors—largely driven by a desire for improved work–life balance [23].

Clearly, the evidence is beyond justifying the potential value of medical FF workplaces. What is needed is to adapt these concepts to medical systems, which are often highly idiosyncratic and diverse workplaces with often-competing needs of doctors, their families, and most importantly patients, at stake. We aimed to use the Delphi consensus methodology to: (i) operationalise what is a “Family-Friendly Culture” (FFC) in a medical system and (ii) to develop a Family-Friendly Self-Audit tool for medical workplaces.

## 2. Materials and Methods

We used Junger et al.’s [24] standard for Conducting and Reporting of Delphi Studies (CREDES) to reach a consensus on operationalised criteria and audit items for Family-Friendly medical cultures. The primary purpose of the Delphi technique is the formation of consensus or the exploration of a field beyond existing knowledge or current conceptual formulations. It has methodological features which enable the involvement of experts with diverse backgrounds irrespective of their geographical location. These features require that: (1) expert ‘panellists’ are questioned about the issue of interest; (2) the process is iterative in nature, comprising several rounds of enquiry; (3) the survey is anonymous to avoid “bandwagon effects” of social conformity and pressure; and (4) the design of each round is informed by a summary of the group response of the previous round. 

According to the aforementioned CREDES guidance on rigour for design and reporting of Delphi studies, prominent criteria for definition, selection, and identification of experts include: representation of a particular profession or stakeholder group; relevant clinical and/or academic expertise; and being a recognized authority in the particular field [24]. Our expert panel was deliberatively recruited to capture a breadth of professional and personal expertise, with a wide range of academic expertise, lived experience of dual commitments to work and family, diversity of family contexts, age and life stage, work settings, and positions. Panellists were all doctors, all still working. Psychiatrists were deliberatively over-recruited based on their clinical and academic subject-matter expertise regarding attachment (hence the psychotherapists), family and system processes (hence family and system therapists), and life cycle stages (e.g., old age psychiatrists). Senior physicians were chosen for their practical expertise in implementation and leadership.

Of the fifteen doctors approached, three male doctors (one male psychiatry trainee; one male psychiatrist and one male paediatrician) although expressing initial interest, declined to participate. The remaining twelve doctors who participated included eight senior psychiatrists (including psychotherapists, family and systems therapists, old age psychiatrists, general adult psychiatrists, a director of psychiatry training and a psychiatry registrar in training). Two panellists were senior physicians of various specialities including a hospital director and a physician head of department; one panellist was a radiologist, and one a surgeon. Age range was 35–81 years old. All worked in public hospital settings, except for two private psychiatrists; and gender distribution included female (n = 11) and male (n = 1). All but one panellist (who was from the United Kingdom) (UK) were working and living in Australia. One panellist identified as having a First Nations background and four panellists identified as having culturally diverse backgrounds.

Seven rounds were conducted. In Round One, panellists were given the frame of the project and the rationale for their nomination as experts in understanding and advocating for doctors’ families and changing health systems to be more “family-friendly”, with an open prompt to brainstorm around this. The other focus of the first round was to define both “family” in this context and “medical systems” or “medical stakeholder cultures.” Using the aforementioned existing FF models [18] as stems, subsequent rounds (two–four) focused on the criteria for Family-Friendly medical cultures, giving free rein to panellists to be aspirational in the elaboration of these criteria until data saturation was reached. In contrast, in Rounds Five to Seven, panellists were asked to generate or modify practical, implementable audit items. 

Depending on availability, panellists were either individually interviewed by the first author (CP) or asked to comment via email on a template document, iteratively developed with each round. Relevant literature was also generated by the consensus group in response to various rounds. The first author (CP) analysed and incorporated the collected comments using content analysis, but more broadly using a thematic analysis approach. Consensus was defined by agreement of ≥75% of the panellists. Upon starting each subsequent round, participants received a summary of anonymous results from the previous round and an updated draft of the document.

## 3. Results

### 3.1. Definitions

We defined “family” in this context as inclusive of all close and intimate relationships, that is, inclusive of the doctor’s intimate partner relationships, relationships with children (including both dependent and adult children), grandchildren, siblings, and parents, as well as all non-traditional family and kinship arrangements. It was also acknowledged that pets are often considered members of the family. 

Accordingly, FF Medical Workplaces allow the doctor to maintain and contribute to their relationships and family caregiving responsibilities by meeting (i) the care and relational needs of family members (including, but not limited to, vulnerable or dependent family members) and (ii) the doctor’s needs to provide care and maintain relationships, and providing for their own needs for affection, care, and support. We noted that dependency and vulnerability determine the way in which family relationships are responded to and prioritised in the workplace. Further to the issue of defining families, the “visible” versus the “invisible” family was identified as a potential source of vulnerability in the workplace. Variations in visibility could be seen in the part-time working parent, the full-time working parent, and the full-time working adult child of ageing parents. Grandparents, women planning pregnancy, breastfeeding mothers, and impending or new fathers can be similarly invisible. 

With regards to defining “culture” and “medical systems” or stakeholders for targeting change, we noted that in medicine there are multiple stakeholders and also a career and family life-cycle time continuum which impacts both the individual’s needs and their ability to access FF practices. Accordingly, we identified potential target systems for change as inclusive of collective employment and training settings such as public hospitals, private hospitals, community services, GP practices, medical colleges, health departments, training networks, and university settings. 

Panellists considered where such change and implementation should be directed. In medical systems, doctors turn first to their line managers or department heads to enquire about family accommodations. As the front-line administrators for each department, it is the obligation and responsibility of department heads (HODs) to be both knowledgeable about FF policies and practices and to actualise them. This is where culture change must be targeted [25]. Any strata higher, and the needs of the organisation will trump those of the medical employees. 

### 3.2. Operationalised Criteria and the Audit Tool

Our operationalised criteria for FF Medical Workplaces were based on the themes that emerged from the Delphi consensus. These criteria, together with illustrative comments and quotes from panellists as well as potentially actionable items are presented in Table 1.

We noted that many of these criteria were aspirational and constrained by the limits of the system, only implementable through changes to the system, not simply Heads of Department (HODs) on their own. Mindful of not adding extra burden to HODS [25] to “fix the unfixable”, the Audit Tool, the FAM-MED, focused on practical, realistic, and implementable FF workplace practices for medical systems (Table 2). With the purposeful aim of disseminating the results of the Delphi consensus to raise awareness and increase understanding of the FF workplace concept within healthcare, we deliberatively made the Audit Tool items somewhat didactic, to encourage learning and improvement with the very act of completing the tool.

## 4. Discussion

We have operationalised the Family-Friendly medical workplace and developed a Family-Friendly Self-Audit tool (FAM-MED) for medical systems to rate their culture with a view to improve them. In doing so, we have raised awareness about the inclusive and dynamic nature of the doctor’s family and the need to adopt a family life cycle approach to FF workplaces [26]. Recommended processes for implementation and sustainability include flexibility and openness to dialogue and feedback, which are key to meeting the individualised but changing needs of doctors while maintaining commitment to optimal patient care. Holding firms to zero discrimination is also integral. We also hypothesise that the Department Head may be the key to implementation. 

We are optimistic that our findings will impart hope in what is often a nihilistic, hopeless medical culture. The fact that we generated 18 items suggests huge potential for systemic movement and change. Notwithstanding its mammoth challenges, the COVID pandemic forced us to adapt our workplaces and, in doing so, has shown that some of the hitherto deemed “outrageous” or “impossible” solutions proposed here are in fact possible. Most pointedly, the COVID pandemic has shown us that working from home has worked very well in many areas and is most adaptable to accommodating work and family in a balanced way. There are many ways in which medicine can incorporate these ‘hybrid’ models, as illustrated by some of our proposals. This is so for all genders and all stages of the life cycle. Dovetailing this is the potential for more flexible options for working arrangements for men, previously dismissed when proposed some 26 years ago [27] but now more acceptable given new role changes for men. Specifically, part-time work, once considered only acceptable for women [27] is now in increasing demand amongst physicians more broadly [28] and is associated with better patient satisfaction and patient experience [29,30].

### Limitations

Firstly, the results of this Delphi consensus must be firmly placed in, and limited to, its Western medical cultural context. While the bidirectional effects of doctor and family well-being are ubiquitous, as evidenced by a study of Thai family medicine residents demonstrating an odds ratio of three for burnout associated with family problems [31], the implementation of the FF workplace must be culturally responsive. What works in Australia, the UK or the United States may not work in Thailand, China or India. However, the rationale is the same, and what this piece of work can do is establish a precedence for working towards culturally appropriate FF Medical Workplaces. 

Secondly, the composition of the Delphi panel may have conferred some limitations. Although deliberative, the over-representation of psychiatrists may inadvertently serve to alienate the medical sector, who may perceive the panel as unrepresentative of the sector as a whole. That said, this limitation may be purely theoretical, as many of the issues at stake transcend speciality craft groups, as was evidenced by the acceptability of the tool when presented to a pilot group of frontline medical staff. 

Moreover, we barely represented the male voice, exacerbating a gaping hole in the literature, where searches for “the physician father” yield little beyond Socrates. This urgently needs remedying with an imperative for future studies to capture the myriad of male physician family experiences. Despite radical changes in workplace participation and family roles over the last century, social constructs of caregiving as a female issue persist. This inequity harms both women and men. The absence of the voices of the men who declined to participate is a salient limitation of our study. The male voice is essential to the design and implementation of family-friendly workplace practices in a way that fosters gender equality both in workforce participation and opportunities and in the opportunity to take up caregiving roles outside of the workplace for both genders. 

We also failed to privilege the voices of non-traditional physician families. Further, the voices of Junior Medical Officers and Doctors in training are also under-represented. Doctors in training face additional challenges in needing to navigate often inflexible training requirements, time-limited contracts, and fierce competition for training positions and jobs. The need to balance these competing demands during this stage of the family life cycle, constrained by the limited chronological window of fertility as well as a range of relational factors, is also acknowledged. 

Thirdly, panellists noted the overlap between FF and “person-friendly” (or person-centred) workplaces and that implicit to creating a FF workplace was a workplace that fostered individual well-being and met individual needs for health promotion, rest, leisure, and connection. It goes without saying that a happy doctor means a happy family, although defining a person-friendly workplace was beyond the scope of this project. 

Fourthly, we do not advocate for an elitist, “For Doctors Only” approach, emphasising the need for FF workplace across the entirety of healthcare, but recognise the particular needs and vulnerabilities of doctors and have developed a bespoke model suited to the medical workplace. 

Finally, we concede that a range of internal (within the doctor) and extrinsic cultural factors limit the implementation of many of these aspirations. Internal barriers include, on the one hand, self-stigma and disempowerment and, on the other hand, the myth of omnipotence and the denial of vulnerability commonly held by doctors. Extrinsic cultural factors include: (i) societal expectations of care and the echoed notion of the omnipotent doctor [32]; and (ii) broader medical culture aspects such as the conveyor belt of training, the sparseness of FF leadership and mentorship, and, finally, resistance from large, insatiable bureaucratic medical systems that favour the organisation over the needs of the staff. Perhaps the most significant extrinsic cultural barrier to the implementation of FF, which relies on goodwill, mutual support, and teamwork, is the milieu of medical culture which has been beset with conflict and intra-team dysfunction for some time [33]. Within an under-resourced and traumatised system, care and compassion will be reduced or chaotic or suffer from rigidity. 

Furthermore, our advocacy for awareness raising and enactment of FF workplace policies is based on an assumption of the existence and adequacy of such policies. While that is true of many medical settings, it is not universally true. 

We recognise that the achievement of these aspirational goals is contingent upon considerable systemic shifts. It is one thing to identify heads of department as ideal targets for implementation of FFMW, it is another thing to ensure that they actually do so. While we have received—in principle—unanimous support for the goals espoused here when tested with a pilot group of medical heads of department in our public healthcare setting, this was likely a self-selected engaged group. The next steps, beyond mere acceptance in principle from an engaged group, are to (i) consider how to engage those less sympathetic to the goal of FFMWs and (ii) test the Audit tool and whether its use improves FF practices. Are our hopes justified? Are the aforementioned obstacles insurmountable? 

We would argue that the doctor, their family, the medical system, and, most importantly, patients gain from FF workplaces in medicine. Few of us can focus 100% on patient care when we know this is at the expense of our families. More generally, it is known that work–family conflict contributes negatively to both life satisfaction and work engagement [34] while FF benefits have a positive effect on organizational performance [35,36]. As previously noted, the toll hospital medicine places on lifestyle and family is often unbearable, and people are exiting medicine [8,23]. Notably, impact on pregnancy, childbearing, childrearing, and relationships and the unavailability of or the questioning of the validity of reasons for requested leave were all factors identified as reasons for women leaving surgery training [37]. We are losing valuable assets; perhaps the very people who prioritise their families make the best doctors. Moreover, while being a doctor and a carer brings with it risks to mental health and well-being by way of exhaustion and anxiety [38], it may also bring a valuable perspective to one’s work with patients and their carers if the system will accommodate us and our needs. Clearly, these are not unsubstantiated hypotheses given the aforementioned increased patient satisfaction and improved patient experience associated with doctors who work part-time [29,30]. We also recognise that, while in the longer term, more FF workplaces may have a positive impact by improving retention and attraction of medical staff, the acute and worsening problem of staff shortages in most countries means that the path to getting there has serious practical challenges.

## 5. Conclusions

Regardless of these challenges, we must persist. We have to stop grooming our trainees for a profession with work–life conflict [38] and self and family sacrifice. It is time to end the martyrdom of medicine. It is time we acknowledge that doctors have families and that what happens at home does not stay at home, and what happens at work does not stay at work. It is time to narrow the gap between identifying as a partner, mother, father, daughter, son, and identifying as a doctor [36]. We affirm the right to be both good doctors and good family members. 

## Figures and Tables

**Table 1 healthcare-11-01679-t001:** What is a Family-Friendly Medical Workplace (FFMW) and how might we measure it?

Operationalised Criteria: Themes	Comments and Quotes	Potentially Implementable FF Practices
**1. Knowledge and awareness**	Leaders and staff know and use the FF policies. They need to be regarded positively rather than putting barriers in the way of doctors who want to use them. Awareness of the legal rights and responsibilities of employers and employees.	HODs * and all current and incoming staff know about available FF policies and resources, understand their responsibilities for supporting FFMW. Use of policies audited.
**2. Not set-and-forget FFMW policy.**	Regular review and assessment of departmental FF policy and practice, informed by staff feedback (Bottom-up, not top-down approach)	Annual Review of Policy. Feedback on staff experiences of FF practices is proactively sought by HOD and responded to. Staff Performance Review used as opportunity for feedback about FF practices.
**3. Advertising and promoting FFMW policy**	Advertising the department FF mission. Being clear that being FF is a priority and goal. Promoting the department as a place where doctors with family caregiving responsibilities can thrive and successfully integrate work and family needs.	Undertaken with all existing and new employees at orientation.
**4. Inclusive and diverse definition of family.**	The inclusive and diverse definition of family must be understood and acknowledged, while guaranteeing respect and safety around privacy. No hierarchy around “who is more deserving”.	Awareness of inclusive and diverse definition of family.
**5. Part-time work and training supported.**	Ensure that part-time positions are of sufficient variety and opportunity compared with full-time terms. No discrimination against part-time workers being less committed, or less part of the department. Recognition of dynamic nature of family caregiving and allow renegotiation of working hours/FTE ^#^ at times around Sentinel Events in the family life cycle—see below.	Offer of part-time work is mandated. KPIs ** of number, percentage, and range of part-time positions of equal variety and opportunity. System requirements of minimal FTE ^#^ are challenged where possible.
**6. FACS, *** maternity and paternity leave supported.**	FACS, *** maternity and paternity leave are encouraged and supported, without career or training repercussions of accessing such. Obliteration of discriminatory hiring practises including veiled discrimination implicit in unanswerable questions such as: “Can you think of any reason you may not be able to fulfil your obligations?”	Flexibility around use of personal leave, sick leave and FACS *** leave. Bureaucratic hurdles to taking FACS and maternity leave are addressed. Leave cover to support such is intrinsic to the organisation.
**7. A mutually respectful arrangement between staff and HOD**	FFMW involves a mutually respectful arrangement involving flexibility and commitment from all parties including the doctor, their colleagues, and the HOD. Notwithstanding unpredictable family emergencies and major medical events (see below), the doctor understands that patient and departmental needs and service requirements must take precedence. We have a service to run or deliver and there are limits to acquiescing to all requests.	Is there an open and respectful process of dialogue between the HOD and the doctor about these mutual responsibilities?
**8. Structural and systemic changes to enable FFMW**	Structural and systemic changes such as leave cover to enable workers to work and have security that when needed (e.g., family emergencies and illness), their family can take precedence. Systemic changes to resourcing (e.g., leave cover) that allow access to meeting these needs. Ideally, such resourcing should be considered and embedded at the time of establishment of departments and reviewed regularly. Doctors are not frightened to disclose family needs (e.g., pregnancy and need for maternity leave and caring needs) for fear of service gaps and impacts on colleagues.	Organizations should have adequate numbers of staff to provide cover for people needing to take leave. Recruitment processes should aim to appoint qualified candidates to the role while addressing diversity imbalances within a department.
**9. Understanding vulnerability and special FF needs of trainee doctors.**	Trainee doctors need to fulfil often competing demands of family, employer, and the College to which they are affiliated at a significant time in the family life cycle (see below). The job, FTE, and leave criteria for both employer and College are often different, resulting in use of fewer FF work options. Employment is often contingent on passing exams that are inflexibly set, sometimes once a year, with an “all-or-nothing” pass barrier.	Flexibility around exam timings should be offered by respective Colleges (e.g., multiple sittings per year, ability to sit exams at multiple stages of training). Sufficient staffing levels to allow medical staff to take study leave. Departments should view rostering practices to ensure that safe work conditions are met and prioritise access to leave for trainee education.
**10. Maintenance of work–family/personal life boundaries**	Work schedules are neither set up to interfere with family life nor scheduled to transgress boundaries of the working day (notwithstanding the fact that this will inevitably happen at times in acute health settings).	Work schedules are FF (meetings, teaching time, etc). Ward rounds are held within the working day.
**11. Zero tolerance for discrimination**	Zero tolerance for discriminatory disparaging comments and behaviours. Bystander effect is problematic in a hierarchical medical system.	Call out and make it clear that all hostile comments and behaviours are unacceptable and violate the rules governing professional conduct.
**12. Flexible work arrangements**	Flexible work arrangements (start and finish times) to accommodate FF practice. Adapted workplaces in response to COVID provide templates for business as usual, not restricted to pandemic measures only. At the same time, the use of flexible work arrangements should not exclude the doctor from being a part of the department.	Consider flexible work arrangements such as early start/early finish times to accommodate family needs (e.g., school pick-up responsibilities). Flexible work environments to allow use of videoconferencing for clinical reviews, departmental meetings and teaching.
**13. Sexuality and gender neutrality in FF policies.**	A recognition that persons of all gender identification may have carer responsibilities and do have needs for affection, care, and support and should not face discrimination when asking for FF work practices. No assumptions that females are the primary caregivers. No longer do eyes raise or are responses such as, “*Can’t your wife do it*?” elicited when men ask for FACS or other leave.	Equal weighting and consideration for persons regardless of gender with regards to FF requests. No unexamined assumptions about the gender of primary caregiver/s in the family.
**14. Costs (to the doctor) of accessing FF policies and practices measured and mitigated.**	Costs of accessing FF policies and practices are measured and mitigated, with safety and respect guaranteed in relation to: (i) risks of disclosure; (ii) privacy i.e., addressing the need to justify who your family is, what their disabilities are, and why you may be needed; (iii) active discrimination; (iv) structural inequity; and (v) bureaucratic burden of approvals briefs and statutory declarations “having to jump through 1000 hurdles.”	HODS are conscious of maintaining safety and preserving privacy for staff accessing FFMW policy.
**15. Minimal forced geographical separation or relocation of families.**	Particularly important for training and service stakeholders.	Strategies to minimise family separations. Regional rotations to offer FF accommodation and prioritise access to childcare. Training organisations to adopt a collaborative approach to trainees with family with respect to term allocations.
**16. Recognition of Family Life cycle approach to FFMW and dynamic nature of caregiving in families.**	Sentinel life-cycle events include but are not limited to: Caring for partner; Achieving pregnancy; Perinatal and postnatal periods; Caring for young children less than 12 years old; Caring for adolescent children; Caring for ageing parents; Caring for grandchildren; Caring for siblings; Caring for disabled family members; Mental or physical illness of any family member; Death or serious illness of a pet; Death or trauma of any family member.	HOD is aware of the Family Life Cycle approach to FFMW.
**17. Consideration of implications for the system of FFMW policies**	Implications may include: (i) Proactive hiring leading to discrimination; (ii) Accommodations creating division (“Why does that person get more flexibility?”) or inequitable burden on those without family responsibilities.	Policies and procedures need to be equitable. Local culture of support needs to be fostered to decrease envy and promote asking according to need.
**18. Domestic violence leave is available, respectful of privacy, and genderneutral.**	That domestic violence leave be available and does not require the disclosure of domestic violence to anyone other than the HOD or Human Resource representative. Domestic violence services offered to staff are offered in a gender-neutral way while maintaining the psychological safety of this vulnerable population.	Domestic violence leave is available and equitable, regardless of gender or sexual orientation. Privacy is respected and there is awareness of available supports. Training colleges develop a position statement on domestic violence to advocate for education of clinicians and support of trainees who are victims of family violence.

**Key:** * HOD = head of department; # FTE = Full time equivalent ratio of working hours (maximum 1.0); ** KPIs = Key Performance Indicators; FACS *** Family and carer services leave.

**Table 2 healthcare-11-01679-t002:** FAM-MED: Family-Friendly Medical Workplace Self-Audit Tool.

Criteria	Never	Sometimes or Occasionally	Frequently or Often
1. I am aware of, read, and refer to the FF policies of my organisation.			
2. I advertise and promote the achievable FF policies of my department to the extent that I can. N.B. I do not promote the unachievable			
3. I seek feedback from staff during their performance review about whether our workplace is meeting their FF needs.			
4. I model respect for FF needs.			
5. I model boundaries between work and home by not scheduling meetings outside working hours.			
6. I model boundaries between work and home by not contacting staff after hours, on weekends or on holidays with the exception of emergencies.			
7. I model boundaries between work and home. I turn on my automatic On Leave message on my email when I am on holiday and encourage staff to do so similarly.			
8. I model making the doctor’s family visible by valuing the importance of my own family and welcoming discussion of family within the workplace.			
9. I do not intrude on staff privacy regarding their reasons for family-related leave or other requests. I do not ask staff to account for themselves when they request leave.			
10. I encourage flexible work practices with regard to working hours.			
11. I encourage flexible work practices with regard to working from home if requested by a staff member while still maintaining the functioning of the unit and optimal patient care.			
12. I encourage part-time work where I possibly can within the constraints of my workplace setting.			
13. I am gender-neutral with respect to my consideration of FF needs and give equal weight to all genders with regard to FF-related requests.			
14. I model and enforce zero tolerance for discrimination in my department. I recognise the inclusive and diverse definition of family while guaranteeing respect and safety around privacy. There is no hierarchy around “who is more deserving.”			
15. I understand the Family Life Cycle approach to FFW. I understand that people have different FF needs at different times of their lives.			
16. I encourage an open and respectful process of dialogue between myself and staff about our mutual responsibilities in regard to FFMW and patient care.			

## Data Availability

The data collected for this review are available upon request from the corresponding author.

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
