# Peer review of "Operationalising the Family-Friendly Medical Workplace and the Development of FAM-MED, a Family-Friendly Self-Audit Tool for Medical Systems: A Delphi Consensus"

_healthcare, 2023, doi:10.3390/healthcare11121679_

Round 1

Reviewer 1 Report

This is a fine and clear article on an important issue. There are a few questions about the expert panel which should be answered in the section on Methods, and a few more general questions which the authors might address in the Discussion.

Questions about the expert panel:

- How were the experts selected? Did everyone who was approached agree to participate?

- Psychiatrists are heavily overrepresented in the panel (8 out of 12). How come?

More general questions:

- The article states that a FFMW should make sure that everyone knows about available FF policies. There seems to be an assumption that FF policies exist and are in themselves adequate. Please comment on this.

- The self-audit tool might help to create FFMWs, but this requires that Heads of Department are going to use it. How can they be persuaded to adopt it, including those who are less sympathetic to the goal  of FFMWs?

Author Response

Response to Reviewer 1 Comments

1.1 Reviewer Comment 1: This is a fine and clear article on an important issue. There are a few questions about the expert panel which should be answered in the section on Methods, and a few more general questions which the authors might address in the Discussion.

1.1 Authors’ Response 1: Thank you for these comments. 

1.2 Reviewer Comment 2: Questions about the expert panel:

How were the experts selected? Did everyone who was approached agree to participate?

1.2 Authors’ Response: Thank you for highlighting this. We have now elaborated on the recruitment addressing these very questions:

According to the aforementioned CREDES guidance on rigour for design and reporting of Delphi studies, prominent criteria for definition, selection and identification of experts include: representation of a particular profession or stakeholder group; relevant clinical and/or academic expertise; and being a recognised authority in the particular field [24]. Our expert panel was deliberatively recruited to capture a breadth of professional and personal expertise, with a wide range of academic expertise, lived experience of dual commitments to work and family, diversity of family contexts, age and life stage, work settings and positions.  Panellists were all doctors, all still working. Psychiatrists were deliberatively over-recruited based on their clinical and academic subject-matter expertise regarding attachment (hence the psychotherapists), family and system processes (hence family and system therapists), and life cycle stages (e.g. old age psychiatrists).  Senior physicians were chosen for their practical expertise in implementation and leadership.

Of the 15 doctors approached, three male doctors (one male psychiatry trainee; one male psychiatrist and one male paediatrician) although expressing initial interest, declined to participate. The remaining 12 doctors who participated included ....

We have additionally now further elaborated on our previous comments about the limitations associated with under-representation of male doctors:    

“Despite radical changes in workplace participation and family roles over the last century, social constructs of caregiving as a female issue persist. This inequity harms both women and men. The absence of the voices of the men who declined to participate is a salient limitation of our study. The male voice is essential to the design and implementation of family friendly workplace practices in a way that fosters gender equality both in workforce participation and opportunities and in the opportunity to take up caregiving roles outside of the workplace for both genders. “  

1.3 Reviewer Comment 3: Psychiatrists are heavily overrepresented in the panel (8 out of 12). How come?

1.3 Authors’ Response: Thank you once again for making another important point. We have now justified the deliberative over-representation: According to the aforementioned CREDES guidance on rigour for design and reporting of Delphi studies, prominent criteria for definition, selection and identification of experts include: representation of a particular profession or stakeholder group; relevant clinical and/or academic expertise; and being a recognised authority in the particular field [24].  Psychiatrists were deliberatively over-recruited based on their clinical and academic subject-matter expertise regarding attachment (hence the psychotherapists), family and system processes (hence family and system therapists), and life cycle stages (e.g. old age psychiatrists).  Senior physicians were chosen for their practical expertise in implementation and leadership.  

We have also conceded this now as a potential limitation:   

“The composition of the Delphi panel may have conferred some limitations Although deliberative, the over-representation of psychiatrists may inadvertently serve to alienate the medical sector, who may perceive the panel as unrepresentative of the sector as a whole. That said, this limitation may be purely theoretical, as many of the issues at stake transcend specialty craft groups, as was evidenced by the acceptability of the tool when presented to a pilot group of frontline medical staff.”  

1.4 Reviewer Comment 4: - The article states that a FFMW should make sure that everyone knows about available FF policies. There seems to be an assumption that FF policies exist and are in themselves adequate. Please comment on this.

1.4 Authors Response: We agree and have now included these comments in the Limitations: “Furthermore, our advocacy for awareness raising and enactment of FFFW policies is based on an assumption of the existence and adequacy of such policies. While that is true of many medical settings, it is not universally true.”

1.5 Reviewer Comment 4: - The self-audit tool might help to create FFMWs, but this requires that Heads of Department are going to use it. How can they be persuaded to adopt it, including those who are less sympathetic to the goal of FFMWs?

1.5 Authors Response : This is an important point. We have now included this and elaborated on our limitations with regards to systemic shifts (pages 10-11, lines 303 onwards)

We recognise that the achievement of these aspirational goals is contingent upon considerable systemic shifts. It is one thing to identify heads of department as ideal targets for implementation of FFMW, it is another thing to ensure that they actually do so. While we have received unanimous in principle support for the goals espoused here when tested with a pilot group of medical heads of department in our public healthcare setting, this was likely a self-selected engaged group. The next steps, beyond mere acceptance in principle from an engaged group, are to consider how to engage those less sympathetic to the goal of FFMWs; and to also test the Audit tool and whether its use improves FF practice.

Reviewer 2 Report

The manuscript was very interesting, and the use of the Delphi method seemed thorough (though the interim steps could be described slightly more clearly to allow for replication / scrutiny). What struck me the most was that many of the medical staff involved were psychiatrists. The intended audience seemed to be the general medical sector, and they might feel that the sample is not as representative of the sector as a whole as it might have been. That said, many of the issues seemed to transcend the medical sector and applied to many other sectors, and perhaps this can be discussed slightly more in the discussion. The other issue was that there was a sense of imbalance, with the practical and financial challenges of the proposed family-friendly policies being briefly acknowledged, but not as thoroughly evaluated and integrated as they might have been. In some countries, staff shortages are acute problems. Although in the longer term, more family-friendly policies may have a positive impact on staffing issues because it might attract more people to the medical profession and retention may be higher, the path to getting there is a serious practical challenge. Some slightly longer evaluation of these issues may make the paper more impactful and less one-sided. I found the table of results slightly reminiscent of the format of standard Human Resources documents (perhaps not specific to the medical sector and perhaps not entirely novel), and perhaps some acknowledgement of that could also be useful, e.g. in the Discussion. Overall, I found the paper inspiring, as it raised an important issue, and provided an important perspective on a major issue in healthcare. I think it would be of great interest to the readers of the journal, healthcare professionals, and Human Resources managers.

Author Response

Response to Reviewer 2 Comments

2.1 Comment The manuscript was very interesting, and the use of the Delphi method seemed thorough (though the interim steps could be described slightly more clearly to allow for replication / scrutiny).

2.1 Authors’ Response : Thank you for these comments. We have now described the interim steps more clearly as suggested:

According to the aforementioned CREDES guidance on rigour for design and reporting of Delphi studies, prominent criteria for definition, selection and identification of experts include: representation of a particular profession or stakeholder group; relevant clinical and/or academic expertise; and being a recognised authority in the particular field [24]. Our expert panel was deliberatively recruited to capture a breadth of professional and personal expertise, with a wide range of academic expertise, lived experience of dual commitments to work and family, diversity of family contexts, age and life stage, work settings and positions.  Panellists were all doctors, all still working. Psychiatrists were deliberatively over-recruited based on their clinical and academic subject-matter expertise regarding attachment (hence the psychotherapists), family and system processes (hence family and system therapists), and life cycle stages (e.g. old age psychiatrists).  Senior physicians were chosen for their practical expertise in implementation and leadership.

Of the 15 doctors approached, three male doctors (one male psychiatry trainee; one male psychiatrist and one male paediatrician) although expressing initial interest, declined to participate. The remaining 12 doctors who participated included ....

We have additionally now further elaborated on our previous comments about the limitations associated with under-representation of male doctors:    

“Despite radical changes in workplace participation and family roles over the last century, social constructs of caregiving as a female issue persist. This inequity harms both women and men. The absence of the voices of the men who declined to participate is a salient limitation of our study. The male voice is essential to the design and implementation of family friendly workplace practices in a way that fosters gender equality both in workforce participation and opportunities and in the opportunity to take up caregiving roles outside of the workplace for both genders. “  

2.2 Comment  What struck me the most was that many of the medical staff involved were psychiatrists. The intended audience seemed to be the general medical sector, and they might feel that the sample is not as representative of the sector as a whole as it might have been. That said, many of the issues seemed to transcend the medical sector and applied to many other sectors, and perhaps this can be discussed slightly more in the discussion.

2.2 Authors’ Response : An excellent point and we have now integrated this into our limitations 

 “The composition of the Delphi panel may have conferred some limitations Although deliberative,  the over-representation of psychiatrists may inadvertently serve to alienate the medical sector, who may perceive the panel as unrepresentative of the sector as a whole. That said, this limitation may be purely theoretical, as many of the issues at stake transcend specialty craft groups, as was evidenced by the acceptability of the tool when presented to a pilot group of frontline medical staff.” 

2.3 Comment The other issue was that there was a sense of imbalance, with the practical and financial challenges of the proposed family-friendly policies being briefly acknowledged, but not as thoroughly evaluated and integrated as they might have been. because it might attract more people to the medical profession and retention may be higher, the path to getting there is a serious practical challenge. Some In some countries, staff shortages are acute problems. Although in the longer term, more family-friendly policies may have a positive impact on staffing issues slightly longer evaluation of these issues may make the paper more impactful and less one-sided.

2.3 Authors’ Response

We agree and have added some of these points, while trying to restore hope:.

We also recognise that while in the longer term, more FF workplaces may have a positive impact by improving retention and attraction of medical staff, the acute and worsening problem of staff shortages in most countries means that the path to getting there has serious practical challenges.

5. Conclusions

Regardless of these challenges, we must persist. We have to stop grooming our trainees....

More thorough evaluation and integration of these challenges is both beyond the scope of this paper and we fear may make it imbalanced in a hopeless, nihilistic direction, the exact opposite of our intent. At the moment, much of the Discussion is spent exploring the challenges.   

2.4 I found the table of results slightly reminiscent of the format of standard Human Resources documents (perhaps not specific to the medical sector and perhaps not entirely novel), and perhaps some acknowledgement of that could also be useful, e.g. in the Discussion.

2.4 Authors’ Response

We concede that format of the results and the Table are very reminiscent of standard Human Resources documents (perhaps not specific to the medical sector and perhaps not entirely novel), but for the same reason as above we feel that pointing out more challenges may be counterproductive towards our intent.  

2.5 Overall, I found the paper inspiring, as it raised an important issue, and provided an important perspective on a major issue in healthcare. I think it would be of great interest to the readers of the journal, healthcare professionals, and Human Resources managers.

2.5 Authors Response: We thank you for these comments.